# Deletion of Acid-Sensing Ion Channel 3 Relieves the Late Phase of Neuropathic Pain by Preventing Neuron Degeneration and Promoting Neuron Repair

**DOI:** 10.3390/cells9112355

**Published:** 2020-10-26

**Authors:** Chia-Chi Kung, Yi-Chu Huang, Ting-Yun Hung, Chih-Yu Teng, Tai-Ying Lee, Wei-Hsin Sun

**Affiliations:** 1Department of Life Sciences, National Central University, Zhongli, Taoyuan City 32001, Taiwan; A00129@mail.fjuh.fju.edu.tw (C.-C.K.); a0928662049@gmail.com (Y.-C.H.); tingyun5903@gmail.com (T.-Y.H.); jerry71726@hotmail.com (C.-Y.T.); tai-ying.lee@wadham.ox.ac.uk (T.-Y.L.); 2Division of Anesthesiology, Fu Jen Catholic University Hospital, New Taipei City 242062, Taiwan; 3School of Medicine, College of Medicine, Fu Jen Catholic University, New Taipei City 242062, Taiwan; 4Department of Physiology, Anatomy and Genetics, University of Oxford, Oxford OX1 3PT, UK; 5Department of Life Sciences and Institute of Genome Sciences, National Yang-Ming University, Taipei 11221, Taiwan

**Keywords:** neuropathic pain, acid-sensing ion channel 3, activating transcription factor 3, M1, M2 macrophages, nociceptors

## Abstract

Neuropathic pain is one type of chronic pain that occurs as a result of a lesion or disease to the somatosensory nervous system. Chronic excessive inflammatory response after nerve injury may contribute to the maintenance of persistent pain. Although the role of inflammatory mediators and cytokines in mediating allodynia and hyperalgesia has been extensively studied, the detailed mechanisms of persistent pain or whether the interactions between neurons, glia and immune cells are essential for maintenance of the chronic state have not been completely elucidated. ASIC3, a voltage-insensitive, proton-gated cation channel, is the most essential pH sensor for pain perception. ASIC3 gene expression is increased in dorsal root ganglion neurons after inflammation and nerve injury and ASIC3 is involved in macrophage maturation. ASIC currents are increased after nerve injury. However, whether prolonged hyperalgesia induced by the nerve injury requires ASIC3 and whether ASIC3 regulates neurons, immune cells or glial cells to modulate neuropathic pain remains unknown. We established a model of chronic constriction injury of the sciatic nerve (CCI) in mice. CCI mice showed long-lasting mechanical allodynia and thermal hyperalgesia. CCI also caused long-term inflammation at the sciatic nerve and primary sensory neuron degeneration as well as increased satellite glial expression and ATF3 expression. ASIC3 deficiency shortened mechanical allodynia and attenuated thermal hyperalgesia. ASIC3 gene deletion shifted ATF3 expression from large to small neurons and altered the M1/M2 macrophage ratio, thereby preventing small neuron degeneration and relieved pain.

## 1. Introduction

Neuropathic pain is a complex, chronic pain state that is usually accompanied by a lesion or disease of the somatosensory nervous system. It is generally believed that neuropathic pain is associated with changes in expression and function of receptors, enzymes and voltage-dependent ion channels in peripheral nerves and dorsal root ganglion (DRG) neurons as well as synapses in the nociceptive pathway in the central nervous system [1,2].

Degeneration of the peripheral nerve causes alterations of Schwann cells that induce a release of tumor necrosis factor α (TNFα), interleukin 1α, β and nerve growth factor (NGF) [3,4,5]. These factors activate resident immune cells (such as microglia, mast cells) to release numerous cytokines and chemokines and thereby recruit immune cells (such as neutrophils, macrophages, T cells) and promote peripheral sensitization of nociceptors [6,7,8]. Chronic excessive inflammatory response in the nervous system after injury may contribute to the maintenance of persistent pain [9]. Macrophage plasticity takes a great role in the chronic inflammatory response and tissue repair [10,11,12]. Classically activated M1 macrophages release pro-inflammatory cytokines to promote inflammatory responses, whereas alternatively activated M2 seems to downregulate inflammation and injury-caused hypersensitivity and promote regenerative growth response [12,13,14,15].

Signaling between the sensory and immune systems is bi-directional. Injured neurons may communicate with their surrounding satellite glia cells (SGCs) via bidirectional release of ATP [16] to release factors (such as TNFα) that activate resident immune cells and recruit more immune cells from the circulation [11,17,18]. They act together to alter the transduction and transmission of nociceptive information by sensory neurons at the level of their target fields, the nerve trunk, DRG and dorsal horn of the spinal cord [7,8,9]. Nerve injury also induces the expression of activating transcription factor 3 (ATF3), a member of the activating transcription factor (ATF)/cAMP-responsive element binding protein (CREB) family of transcription factors, in DRG neurons in a distance-dependent manner [19], but its expression is downregulated by NGF [20]. ATF3 seems upregulated in regenerating neurons, in association with axon sprouting and growth [21]. It probably prevents neuronal death by increasing heat shock protein 27 (HSP27) expression [22].

Although the role of neurotropic factors and cytokines in mediating allodynia and hyperalgesia has been extensively studied, the detailed mechanisms or responding time between neurons and glia, glia and immune cells or between neurons and immune cells have not been completely elucidated. Acid-sensing ion channels (ASICs) are voltage-insensitive, proton-gated cation channels that are activated by extracellular acidic pH. Among the ASIC subunits, ASIC3 is the most essential pH sensor for pain perception [23,24]. ASIC3 gene expression is increased in DRG neurons after inflammation and nerve injury and is involved in macrophage maturation [25,26,27]. Blocking ASIC3 function by selective antagonist APETx2 significantly inhibits primary inflammatory thermal hyperalgesia at the peripheral level [23]. Interestingly, ASIC3-deficient mice show reduced secondary mechanical hyperalgesia, rather than primary hyperalgesia, in the joint inflammation model, suggesting that ASIC3 activation at peripheral afferents sends nociceptive signals to the central nervous system to result in central sensitization [28]. In dual acid injection model, Chen et al. [29] found that ASIC3 enhances activity of voltage-gated sodium channel 1.8 (Na_v_1.8) to regulate the transition from acute to chronic muscle pain. The similar mechanism may be applied in modulation of ASIC3 in the chronic phase of CFA-induced RA model [30]. However, whether prolonged hyperalgesia induced by nerve injury also requires ASIC3 remains unclear.

In this study, we established a model of chronic constriction injury (CCI) of the sciatic nerve in ICR mice. CCI mice showed long-lasting mechanical allodynia and thermal hyperalgesia. CCI also caused long-term inflammation at the sciatic nerve, increased satellite glial expression, primary sensory neuron degeneration and increased ATF3 expression. ASIC3 deficiency shortened mechanical allodynia and delayed and attenuated thermal hyperalgesia. ASIC3 gene deletion also caused a shift of ATF3 expression from large to small neurons and altered the M1/M2 macrophage ratio, thus preventing small neuron degeneration and relieved pain.

## 2. Materials and Methods

### 2.1. Animals

Male and female ICR mice (8–12 weeks old) were purchased from BioLASCO Taiwan (Taipei, Taiwan) and housed 3–4 per cage under a 12-h light/dark cycle (lights on at 7:00) with food and water ad libitum in a temperature- and humidity-controlled environment at the National Central University. Care and use of mice conformed to the Guide for the Use of Laboratory Animals (US National Research Council) and the experimental procedures were approved by the local animal use committee (IACUC, National Central University, Taiwan). All behavioral testing was performed between 9:00 and 17:00. Efforts were made to minimize the number of animals used and their suffering. For gene expression, mice were placed in the euthanasia chamber and killed by introducing 100% CO_2_ with a fill rate of 20% to 30%/min. Mice were unconscious usually within 2 to 3 min. DRG were excised for RNA extraction.

ASIC3^−/−^ and ASIC3^+/+^ mice (on an ICR background) were generated as described [31]. The genotyping primer sequences were for ASIC3^−/−^, 5′-attcaggctgcgcaactgtt/5′-tgtggtcccaggacttggta; and ASIC3^+/+^, 5′-cacagctccaggaggagttgaa/5′-ccttgtgacgaggtaacaggta.

### 2.2. Surgery

For the CCI model, mice were anesthetized with avertin (0.4–0.6 mg/gm, i.p., Sigma-Aldrich, Saint Louis, MO, USA) and the right sciatic nerve was exposed at the mid-thigh level. Three ligatures (3-0, silk, Unik, Taiwan) were tied loosely around the nerve at about 1 mm apart. In sham-operated mice, the same procedure was performed but without ligating the nerve.

After surgery, mice underwent behavioral tests or were killed at specified weeks to collect the sciatic nerve or DRG. The sciatic nerve was excised and observed under a dissecting microscope (Leica EZ-4 stereo microscope, Bensheim, Germany) or fixed for histological staining. In some experiments, lumbar 4, 5 DRG were isolated after CCI surgery for measuring gene expression or immunostaining.

### 2.3. Histological Staining and Immunostaining

At 0, 1, 2, 3, 4, 8 and 14 weeks after CCI surgery, the sciatic nerve from the ipsilateral or contralateral side was excised, fixed in 25% formalin overnight and then transferred to 50% ethanol. Fixed tissues were embedded in paraffin and sectioned with use of a microtome, then stained with hematoxylin and eosin (by the Taiwan Mouse Clinic, Taipei, Taiwan) or with anti-CD80 or anti-CD163 antibodies. For CD80 or CD163 staining, sections were incubated with 100% Xylene for 10 min twice, and then sequentially incubated for 5 min once with 100%, 95%, 90%, 80% and 70% alcohol. After a 5-min washing with water, sections were incubated with TBS-TX (Tris-buffered saline containing 0.1% Triton-X100) for 2 min, then blocked with TBS containing 1% BSA for 2 h. Sections were incubated with anti-CD80 (1:250, Biorbyt, Cambridge, UK) or anti-CD163 (1:100, Biorbyt, Cambridge, UK) primary antibodies overnight, then with alkaline phosphatase-conjugated anti-rabbit IgG antibodies (1:1000) for 1 h, followed by signal development with 5-bromo-4-chloro-3-indolyl phosphate/nitro blue tetrazolium (BCIP/NBT, Sigma-Aldrich, Saint Louis, MO, USA) for 1 h. Sections were then washed with water and incubated for 15 s with 95%, and 100% alcohol for 5 min with 100% Xylene before mounting with Organo/Limonene Mount (Sigma-Aldrich, Saint Louis, MO, USA).

Immune cells were observed under a light microscope with a 10× or 100× objective (Leica DM500, Bensheim, Germany) and the number of cells was counted in a 1-mm^2^ region and represented as cell density (cells/mm^2^). Each slide contained two sections and 5–6 regions were selected for each section.

For immunostaining, L4 or L5 DRG tissues were excised at 0, 1, 2, 3, 4, 5, 7, 8 and 14 weeks after CCI surgery and immediately placed into freezing solution. Serial sections 12 μm thick were cut by use of a cryostat (Leica microsystem 3510S, Bensheim, Germany). Sections were stained with anti-peripherin (PERI, 1:500, Sigma-Aldrich, Saint Louis, MO, USA), anti-N52 (N52, 1:500, Sigma-Aldrich, Saint Louis, MO, USA), anti-glial fibrillary acidic protein (GFAP; 1:1000; Dako, Santa Clara, CA, USA), or anti-ATF3 (1:200, Santa Cruz, Houston, TX, USA) primary antibodies, followed by FITC-conjugated goat-anti-rabbit-IgG antibody (1:250, Sigma-Aldrich, Saint Louis, MO, USA) or TRITC-conjugated goat–anti-mouse IgG antibody (1:250, Sigma-Aldrich, Saint Louis, MO, USA). All antibodies were diluted in 1× PBS containing 1% bovine serum albumin. All antibody incubations were carried out at 4 °C overnight. The specimens were examined by using a 63× objective on a confocal microscope (Zeiss, LSM 700, Oberkochen, Germany) or a fluorescence microscope (Leica DMI3000B, Bensheim, Germany). For confocal microscopy, the digitized images were captured by using AxioVixion and the fluorescence intensity was quantified by using Image J (National Institutes of Health, Bethesda, MD, USA). For fluorescence microscopy, PERI-positive neurons surrounded by GFAP-positive SGCs in one-third or more of the PERI circumference were counted (PERI^GFAP+^) and expressed as a percentage of total PERI-immunoreactive (IR) neurons (PERI_T_) in the fields analyzed. Data for each treatment group were collected from 7 to 13 slides, and each slide contained more than 8 DRG sections.

### 2.4. Tests

To assess mechanical nociceptive responses, animals were tested for withdrawal thresholds to mechanical stimuli (von Frey filaments, Touch-Test, North Coast Medical, Morgan Hill, CA, USA) applied to the plantar aspect of the hind paw. Mice (n ≥ 6 per group) were pre-trained for 1 to 2 h each day for 2 days before the test as previously described [32]. Von Frey fibers were applied 5 times at 5-s intervals to the plantar surface of each hind paw at various times after injections. The paw withdrawal threshold (PWT) was determined when paw withdrawal was observed in more than 3 of 5 applications.

Animals were also tested for thermal nociceptive response to radiant heat applied to the plantar surface of the paw. After pre-training in transparent plexiglass chambers (10 × 8 × 10 cm/chamber) on a glass floor, the plantar surface of mouse hind paws was stimulated with a light bulb (30% intensity, 251 mW/cm^2^). The latency to withdrawal of the paw (PWL) from radiant heat was measured. Measurements from three trials at 1-min intervals in each paw were averaged.

### 2.5. Statistical Analysis

All data are presented as mean ± SEM. One- or two-way ANOVA with post-hoc Bonferroni test was used to compare results for multiple groups. For estimating population proportion, the z-test was used to test level of significance, with 95% confidence intervals for proportions estimated from ([33] Table 1 of Biometrika tables for statisticians). *p* < 0.05 was considered statistically significant.

## 3. Results

### 3.1. CCI of Sciatic Nerve Causes Long-Term Hyperalgesia, Inflammation, and Neuron Degeneration

The CCI model developed by Bennett and Xie [34] appears to be one of the most frequently used models for studying neuropathic pain and its treatment. To explore mechanisms of neuropathic pain, we established the CCI model in mice by three loose ligatures on the mid-sciatic nerve. Mechanical allodynia was evaluated by PWT with von Frey filaments. Thermal hyperalgesia was evaluated by PWL according to Hargreaves’s method [35]. The baseline PWT and PWL on the ipsilateral side was 3.00 ± 0.37 g (n = 8) and 15.50 ± 0.83 s (n = 7), respectively. Similar to observations by Bennett and Xie [34], CCI mice displayed long-term unilateral mechanical allodynia (Figure 1A) and thermal hyperalgesia (Figure 1B). PWT on ipsilateral side was decreased strongly after nerve injury, to 0.28 ± 0.05 g (n = 8) at week 1, then kept below 1.0 g for at least 14 weeks. The mean PWT at week 1 was 0.28 ± 0.05 and 2.5 ± 0.33 g (n = 8) on the ipsilateral and contralateral site, respectively, with significant difference between both sites (*p* < 0.01) (Figure 1A). Similarly, PWL significantly decreased to 9.40 ± 0.54 s (n = 7) at week 1 and remained lower than 11 s (n = 7) at week 14 (Figure 1B). No significant difference between female and male in mechanical allyodynia was found after CCI surgery (Figure 1C).

After ligatures were removed, sciatic nerves had marked constrictions beneath each of the 3 ligatures in the first week (Figure 1D). The diameter of the sciatic nerve was reduced. Nerves taken 2 weeks post-operatively showed a progressive tendency for adjacent constrictions to merge and a distinct swelling. By week 3, the ligated regions of the nerves were completely encapsulated in a dense mass of connective tissue that adhered to adjacent muscle. Nerves taken 4 weeks after surgery showed constriction or a region of nearly uniform thinning. By 8 weeks after surgery, the connective tissue capsule had been resorbed. The ligated region remained thin but appeared opaque and a dull yellow color. Nerves taken 14 weeks after surgery were thick. The constriction of the sciatic nerve was associated with intraneural edema, focal ischemia and Wallerian degeneration.

To understand which types of neurons were degenerated during CCI, we labeled DRG large- or small-diameter neurons with intermediate filament markers N52 and PERI, respectively, to examine neuron degeneration. After nerve injury, neurons were degenerated, and filament markers had disappeared. The number of DRG neurons with positive filament markers was decreased and reached the maximum decrease (40% of total degenerated) at week 4, and then 13% lost at week 14 (Figure 1E,F). For large-diameter neurons (N52^+^), 11% were degenerated at the first week but reversed back to untreated baseline level from week 5 (Figure 1E,G). However, 15% of small-diameter neurons were degenerated at the first week and further decreased to 60% at week 4. Approximately 20% of small-diameter neurons were still degenerated at 14 weeks (Figure 1H).

Histology examination at the cutting end of the ligation site of sciatic nerves revealed immune cells gathering at the peri-epineurium area at week 1, then remarkable immune cell infiltration into the inner side of nerves after week 2 (Figure 2A). Immune cells can be divided into three types: macrophages, granulocytes and lymphocytes. On examination of immune cell density of the nerve injury site, total immune cells showed significantly increased density from week 2 to 14. Granulocytes showed a significant increase at week 2, lymphocytes at weeks 2 and 3, and macrophages at weeks 2, 3, 4, 8 and 14. Increased macrophage population mainly contributed to the changed total immune cell density (Figure 2B–E).

### 3.2. Number of SGCs and ATF3-Positive Neurons Was Increased after CCI Surgery

Consistent with previous study in spinal nerve ligation [36,37], the expression of GFAP, a marker of glial cells, was gradually increased with time on the ipsilateral side after CCI surgery, with significant expression at weeks 2, 3, 4 and 14 (Figure 3A,B). The proportion of PERI^+^ neurons surrounded by GFAP^+^ SGCs significantly grew during weeks 2, 4, 8 and 14, with a gradual increase from 10% to 25% from week 2 to 14 (Figure 3C).

To understand which types of neurons were injured, we used ATF3 to label injured neurons. Intracellular ATF3 expression was increased significantly after nerve injury. Only some (1–2%) ATF3^+^ neurons co-localized with PERI^+^ neurons (ATF3^+^/PERI^+^), which did not significantly differ between weeks 1 and 14 in PERI^+^ neurons (Figure 4A,B). Most ATF3^+^ neurons were co-localized with N52, and the number of ATF3^+^/N52^+^ neurons gradually increased from week 1 to 14 (Figure 4C,D). Two transition zones of ATF3^+^ neuron distribution were revealed. One zone started from week 2 to 4, with ATF3^+^ expression shifted from 10–20 μm size to 25–35-μm size (Figure 4E). Another zone time interval started from week 5 to 8, with similar ATF3^+^ expression shifting from 15 to 25 μm, to 25 to 35 μm size (Figure 4E). ATF3^+^ expression seemed to be associated with N52^+^ and PERI^+^ neuron regeneration. At weeks 1–4, both N52^+^ and PERI^+^ neuron number decreased over time, then a transition started from week 5, recovering the N52^+^ neuron number to the baseline level and PERI^+^ number to 80% of the control level (Figure 1F,G).

### 3.3. ASIC3 Deficiency Reverses CCI-Induced Hyperalgesia and Delays Neuron Loss in the Beginning of CCI

ASIC3^+/+^ mice displayed long-term mechanical and thermal hyperalgesia after CCI surgery (Figure 5A,B), which is consistent with CCI ICR mice. In ASIC3^−/−^ mice, PWT slowly increased from week 4, then returned to baseline after week 8 (Figure 5A), which suggests that ASIC3 deletion shortened the mechanical hyperalgesia. Unlike in ASIC3^+/+^ mice, PWL in ASIC3^−/−^ mice was not decreased at week 1 but was decreased after week 2 with several up–down fluctuations (Figure 5B). However, PWL was significantly higher in ASIC3^−/−^ than ASIC3^+/+^ mice, which indicates that ASIC3 deletion delayed the onset of thermal hyperalgesia and attenuated thermal hyperalgesia. Similar results were found using ASIC3 selective antagonist APETx2 to block ASIC3 function (Figure 5C,D). APETx2 reduced mechanical allodynia at 8 w after CCI and attenuated thermal hyperalgesia at 4 and 8 w after CCI. These results confirmed that behavioral changes observed in ASIC3^−/−^ mice were due to loss of ASIC3 function, rather than other gene compensation caused by ASIC3 deletion.

Similar to CCI ICR nerves, ASIC3^+/+^ sciatic nerves showed morphological nerve axon strangulation and edematous degenerative changes after untied constriction nodes at week 1 (Figure 5E). We found further degeneration changes such as axonal degeneration during the first 8 weeks and regeneration such as marked axon growth at week 14. However, a mass swelling like an axon sheath appeared at week 8 and 14 in the ASIC3^−/−^ sciatic nerve, which indicates more substantial regeneration.

The DRG neuron degeneration was delayed in the beginning of CCI in ASIC3^−/−^ mice. At week 1, only 3% of N52^+^ and 8% of PERI^+^ neurons were degenerated in ASIC3^−/−^ mice, as compared with 17% and 19%, respectively, in CCI ICR mice (Figure 5F–I), but the neuron degeneration was greatest at week 4. Surprisingly, PERI^+^ neuron number was significant higher in ASIC3^−/−^ than CCI ICR mice at week 8, but the N52^+^ neuron number was reduced at weeks 4 and 8 (Figure 5H,I). Less large and small neuron degeneration at week 1 may be related to the delayed onset of thermal hyperalgesia, whereas less small neuron degeneration at week 8 might contribute to shortening prolonged mechanical hyperalgesia.

### 3.4. ASIC3 Deficiency Reverses the Shift from Large to Small Cells in ATF3^+^ Neurons, with No Alteration in Gliosis

Although ATF3^+^/N52^+^ neurons were also increased with time after CCI, a significant decrease occurred at week 8 in ASIC3^−/−^ mice (Figure 6A,B). This finding may be associated with a decrease in N52^+^ neuron number at week 8 in ASIC3^−/−^ mice (Figure 5F). The distribution of ATF3^+^/N52^+^ expression differed in ASIC3^−/−^ mice. At weeks 1 and 14, the distribution was similar in both ASIC3^−/−^ and wild-type mice: ATF3^+^ neurons were distributed broadly from 10 to 40 μm size with a peak in 20–25 μm size. Nevertheless, at weeks 4 and 8, ATF3^+^ neurons were distributed in the 10–25 μm size in ASIC3^−/−^ mice as compared with the 25–35 μm size in wild-type mice (Figure 6C). Given a shift of ATF3^+^ neurons from large to small neurons in ASIC3^−/−^ mice, deletion of ASIC3 may lead to less injury in small size neurons. The shift may account for less small-neuron degeneration but greater large-neuron degeneration at week 8 (Figure 1F,G) and seemed to be associated with a reversal of mechanical allodynia from week 8 in ASIC3^−/−^ mice (Figure 5A).

Similar to wild-type mice, in both ASIC3^+/+^ and ASIC3^−/−^ mice, the number of SGCs was gradually increased after CCI, with no significant difference between the two groups (Figure 7A,B), which suggests that ASIC3 deficiency does not alter gliosis.

### 3.5. ASIC3 Deletion Alters M1/ M2 Macrophage Ratio after CCI

ASIC3^+/+^ mice showed a similar increasing pattern of total immune cell density as wild-type mice (Figure 8A–E). However, ASIC3^−/−^ mice showed a significant cell density increase and decrease at weeks 1 and weeks 4 and 8, respectively (Figure 8B). Such changes were mainly attributed to changes in macrophage number (Figure 8C–E). Macrophage number was first increased at week 1, then decreased at weeks 4 and 8 in ASIC3^−/−^ mice.

To understand whether the inflammatory response induces a change of macrophage polarization and phenotypes, CD80 and CD163 were used as markers of M1 and M2 macrophages, respectively. The cell density of M1 macrophages was gradually increased from weeks 1 to 14 in both ASIC3^+/+^ and ASIC3^−/−^ mice, with the increase significantly higher in ASIC3^+/+^ mice at weeks 4, 8 and 14 (Figure 9A,B). The density of M2 macrophages was lower at weeks 1 and 4 but was largely increased at week 8 in ASIC3^+/+^ mice. M2 cell density was significantly higher in ASIC3^−/−^ than ASIC3^+/+^ mice from weeks 1 to 14 (Figure 9A,C). The change in M1:M2 ratio between ASIC3^+/+^ and ASIC3^−/−^ mice was substantial. In ASIC3^+/+^ mice, the M1:M2 ratio was 89:11, 91:9, 70:30 and 95:5 at weeks 1, 4, 8 and 14, respectively. In ASIC3^−/−^ mice, the ratio was 72:28, 58:42, 56:44 and 62:38, respectively, with approximately 20% and 30% increase in M2 ratio at weeks 1 and 4, respectively (Figure 9D,E).

## 4. Discussion

In this study, we have demonstrated that CCI induced long-lasting unilateral hyperalgesia for mechanical or thermal stimuli. Chronic hyperalgesia was attributed to (1) long-lasting inflammation at the sciatic nerve: the number of total immune cells was increased over time with increased granulocyte number at 2 weeks and macrophage number starting at 2 weeks; (2) nerve degeneration: both large and small-diameter neurons were degenerated and ATF3 expression in large-diameter neurons was increased with time to prevent large-neuron degeneration at the later time; (3) increased number of SGCs: the number of SGCs surrounded by small-diameter neurons was increased with time. In addition, ASIC3 gene deletion significantly shortened mechanical allodynia and attenuated thermal hyperalgesia. ASIC3 deficiency (1) altered the M1/M2 macrophage ratio by decreasing M1 number but increasing M2 number and (2) reduced small-diameter neuron degeneration and shifted ATF3 expression from large- to small-size neurons, preventing small neuron degeneration. Less small-neuron degeneration and a shift in M1/M2 ratio contributed to shortening mechanical allodynia and attenuating thermal hyperalgesia.

Our CCI model, consistent with the previous results of Bennet and Xie [34], showed long-term unilateral mechanical and thermal hyperalgesia. After surgery, both large- and small-diameter axons were degenerated. ATF3, belonging to the ATF/CREB family of transcription factors, is notably induced in corresponding tissue that is exposed to the neuron injury or stress signals [38] and associated with a regeneration program [39]. ATF3 prevents neuronal degeneration and promotes neuronal regeneration probably through several ways: (1) in demyelinated neurons, ATF3 which may dimerize with c-Jun increases the speed of axonal mitochondrial transport to maintain axonal homeostasis, preventing neuronal degeneration [40]; (2) in non-injured neurons, ATF3 may promote HSP27 expression to prevent neuronal death [22]; (3) in Schwann cells, ATF3 which may dimerize with c-Jun induces gene expression for injured axon regeneration [41]; (4) in macrophages, ATF3 expression may promote polarization of M2 macrophages which enhance clearance of necrotic debris and promote axonal regeneration [13,42,43]. ATF3 showed low expression in small-diameter neurons: ATF3^+^ neuron number was about 1% to 2% at weeks 1 and 14 after CCI. However, ATF3 expression was strong in large-diameter neurons and increased with time. Approximately 20% of large-diameter neurons were ATF3-positive at week 14 after CCI. These results were consistent with the observation of cell type markers. Actually, 20% of large-diameter neurons were degenerated in the first week after CCI, but such degeneration was reversed back at week 5. Approximately 20% of small-diameter neurons were degenerated in the first week and further increased to 50% at week 4. Most small-diameter neurons did not express ATF3, so a proportion of small neurons (25%) was still degenerated even at week 14 after CCI.

Neuronal degeneration was also observed in previous studies in rat or mice [44,45,46,47]. Degeneration of nerve fibers is relevant to hyperalgesia development [45,48,49]. C57BL/Wld mice with slow Wallerian degeneration rate due to a genetic defect showed delayed thermal hyperalgesia [49]. Macrophages and secreted cytokines affect Wallerian degeneration, influencing hyperalgesia development [48]. Coggeshall et al. [50] suggested that loss of large fibers is associated with the onset of hyperalgesia, but the subsequent course of hyperalgesia is not related to loss of large fibers. Our results were consistent with the previous study by Coggeshall et al. [50]. In our case, both large and small fibers were degenerated from the first week, corresponding to the onset of mechanical and thermal hyperalgesia. After week 5, the large-fiber number was recovered, but substantial proportions of small fibers were still degenerated, corresponding to a subsequent course of hyperalgesia. Axon degeneration were attenuated approximately 4 weeks after CCI surgery, followed by regeneration from week 5. Thus, 4 to 5 weeks after CCI may be the transition from degeneration to regeneration.

In ASIC3-deficient mice, axon degeneration was greatly delayed during the first week. Only 8% of small fibers and 3% of large fibers were degenerated as compared with ~20% fibers degenerated in wild-type CCI mice. This result reflects a delay in the onset of thermal hyperalgesia in ASIC3^−/−^ mice, which is consistent with a previous study finding that slower degeneration delays hyperalgesia development [49]. In ASIC3^+/+^ control mice, degenerated axons were regenerated, and most were large fibers. However, most regenerated axons in ASIC3^−/−^ mice were small fibers. A large proportion of small-fiber regeneration at week 8 reflects reversed mechanical allodynia from week 8 in ASIC3^−/−^ mice. ATF3 expression also shifted to small-size neurons (10–30 μm) at week 8 in ASIC3^−/−^ mice, corresponding to small fiber regeneration. Of note, we also found the shift of ATF3 expression to small-size neurons at week 4, with no significant increase in neuron number. The transition from degeneration to regeneration may have started earlier in ASIC3^−/−^ mice, despite no marked increase in fiber number. ASIC3 deficiency may prevent neuron degeneration, especially in small neurons. This situation reflects the shortening of mechanical allodynia and delayed attenuated thermal hyperalgesia caused by ASIC3 deficiency.

The immune cells were increased in number over time after CCI surgery. The number of macrophages was not increased until week 2 in ASIC3^+/+^ mice after CCI, but in ASIC3^−/−^ mice, was increased during the first week after CCI. ASIC3 deficiency may increase the proliferation of resident macrophages (endoneurial macrophages) but not recruited hematogenous macrophages. The number of resident endoneurial macrophages is about 9% of the entire cell population, and they perform myelin phagocytosis earlier than recruited macrophages [7,51]. Phagocytosis and removal of degenerating myelin by both resident and recruited macrophages is essential for successful axonal regeneration. Increased number of endoneurial macrophages at the beginning of injury could be related to protection of small-diameter neurons observed in ASIC3^−/−^ mice. M2 macrophages could enhance clearance of necrotic debris and promote axonal regeneration [13,42] and also have anti-inflammatory and analgesic roles [15]. In ASIC3^+/+^ mice, M2 macrophages were increased in number at week 8 after CCI, which then decreased. Inflammation and pain continued, but a proportion of neurons were regenerated in ASIC3^+/+^ mice in the later phase. The M2 increase may contribute to the regeneration response more than to anti-inflammatory or analgesic effects after week 8. In ASCI3^−/−^ mice, M2 macrophages were greatly increased in number from the first week after CCI, which lasted for 14 weeks. This increase may reflect reduced mechanical allodynia from week 8, attenuated thermal hyperalgesia from week 1, and neuronal regeneration at weeks 1 and 8. ASIC3 deletion may regulate M2 macrophage polarization to promote neuronal regeneration and relieve pain. The chronic mechanical pressure (CCI surgery) may have produced ischemia and compromised delivery of oxygen and nutrients. Acute hypoxia favors M2 polarization, but chronic hypoxia triggers M1 polarization [52]. Thus, the ratio of M1:M2 was kept high (90:10), except for a transient switch to 70:30 at week 8, in ASIC3^+/+^ mice. High ratio of M1:M2 reflects the development of chronic inflammation and pain. Extracellular acidosis is up-regulated phagocytic receptor stabilin-1 to enhance phagocytic activity in JNK (c-Jun N-terminal kinase)-dependent manner [53]. Kong et al. [26] found that ASIC3 non-selective blocker abrogates acid-induced increase of pinocytosis (the ingestion of extracellular fluids), antigen presentation, and IL-10 secretion. It is likely that ASIC3 respond acid signals to increase stabilin-1 expression to regulate phagocytic activity of macrophages. Without the ASIC3 gene, the phagocytic activity of macrophage and response to chronic hypoxia could be inhibited. We found the ratio of M1:M2 lowered to 55:45, which shortened the maintenance of the chronic pain state.

## 5. Conclusions

CCI surgery in mice induced high M1:M2 macrophage ratio, increased SGC number and ATF3 expression and caused neuronal degeneration, resulting in chronic inflammation and pain. ASIC3 gene deletion reduced the M1:M2 ratio and switched ATF3 expression to small neurons, which attenuated neuronal degeneration and relieved pain. Accordingly, ASIC3 may regulate macrophage polarization and ATF3 expression to modulate pain.

## Figures and Tables

**Figure 1 cells-09-02355-f001:**
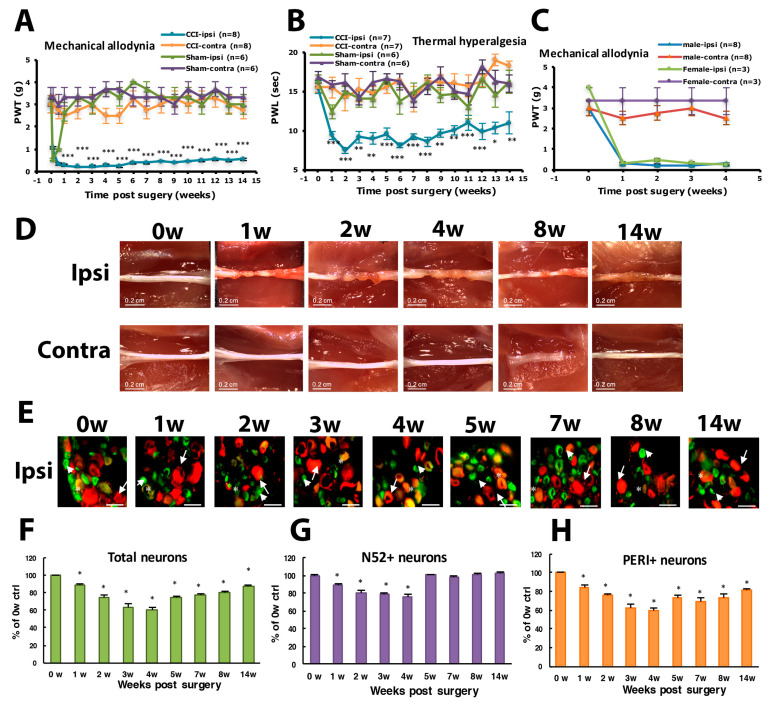
Chronic constriction injury (CCI) of the sciatic nerve in mice induces long-term hyperalgesia and neuron loss. The sciatic nerve of ICR mice was loosely ligated three times (CCI) or was sham operated (Sham). Paw withdrawal threshold (PWT, **A**) and paw withdrawal latency (PWL, **B**) were measured before (0 weeks [w]) or at 1–14 w after surgery. Data are mean ± SEM of PWT (CCI n = 8; Sham n = 6) or PWL (CCI n = 7; Sham n = 6). * *p* < 0.05; ** *p* < 0.01; *** *p* < 0.001 for ipsilateral vs. contralateral side by two-way ANOVA with post-hoc Bonferroni test. (**C**) PWT for male (n = 8) and female mice (n = 3). (**D**) The sciatic nerve from the ipsilateral or contralateral side of CCI mice was excised at 0, 1, 2, 3, 4, 8 and 14 w, and ligature was removed, followed by observation by dissecting microscopy. Scale bar is 0.2 cm. (**E**–**H**) Lumbar 5 dorsal root ganglia (DRG) from the ipsilateral side of CCI mice were taken at 0, 1, 2, 3, 4, 5, 7, 8, 14 w after CCI surgery, then immunostained with anti-peripherin (PERI, green) or anti-N52 (N52, red) antibodies. Cell images are shown in E, and histograms show percentage of total cells (F), N52-positive (N52^+^) neurons (**G**), and PERI-positive (PERI^+^) neurons (**H**). Arrows indicate PERI^+^ neurons and arrowheads N52^+^ neurons. Scale bar is 50 μm. * *p* < 0.05. Each time point contains n = 468–1304 neurons. n = 2 mice.

**Figure 2 cells-09-02355-f002:**
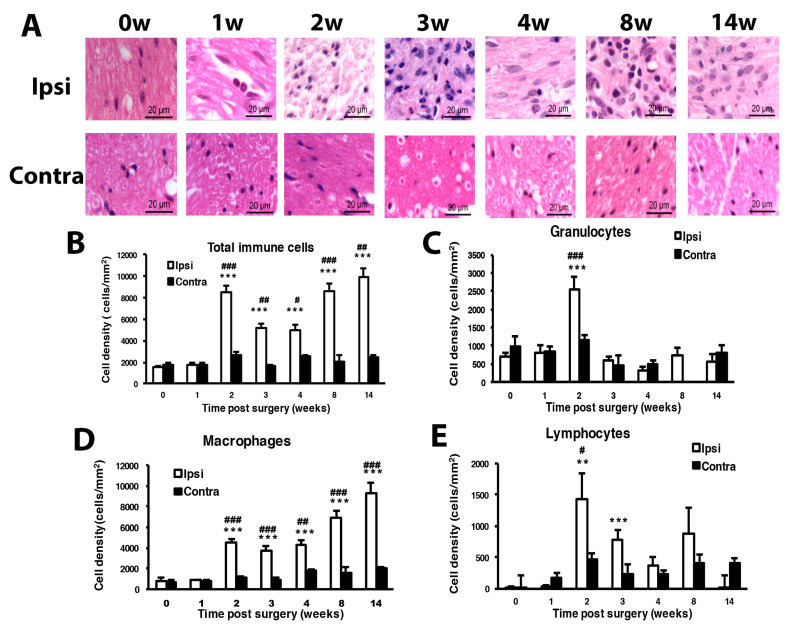
Mice with CCI of the sciatic nerve show long-term inflammation. Sciatic nerves were excised from the ipsilateral or contralateral side of CCI mice at 0, 1, 2, 3, 4, 8 and 14 w, fixed and cross-sectioned, followed by H and E staining. Representative H and E-stained sections showing infiltration of immune cells in the sciatic nerve (**A**). Scale bar is 20 μm. Histograms show the cell density (cells/mm^2^) of total immune cells (**B**), granulocytes (**C**), macrophages (**D**), and lymphocytes (**E**). The cell number was calculated from regions of 1 mm^2^ to obtain cell density (0, 14 w, n = 5; 1 w, n = 4; 2, 3, 4, 8 w, n = 6 mice). # *p* < 0.05; ## *p* < 0.01; ### *p* < 0.001 for ipsilateral vs. contralateral side, and ** *p* < 0.01, *** *p* < 0.001 for different weeks vs. 0 w by two-way ANOVA with post-hoc Bonferroni test.

**Figure 3 cells-09-02355-f003:**
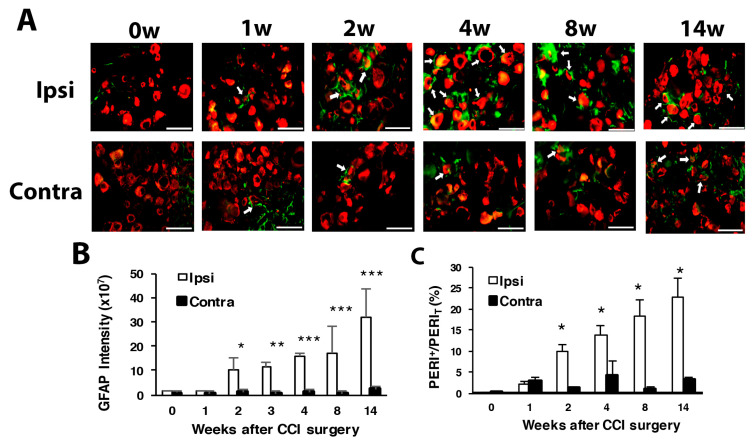
Number of satellite glial cells is increased with time after CCI surgery. The ipsilateral and contralateral L5 DRG were excised at 0, 1, 2, 4, 8 and 14 w after CCI surgery and co-stained with anti-GFAP and anti-PERI antibodies. (**A**) Cell images showing GFAP (green) or PERI (red)-positive neurons. Scale bar is 50 μm. (**B**) GFAP intensity quantified by the confocal microscope. * *p* < 0.05, *** *p* < 0.001 for 0 w vs. other weeks by two-way ANOVA with post-hoc Bonferroni test. (**C**) Percentage of PERI-positive neurons surrounded by GFAP^+^ cells (PERI^GFAP+^) among total PERI-positive (PERI_T_) neurons. Ipsilateral side n = 369–1196; contralateral side n = 196–1307 neurons; n = 2 mice.

**Figure 4 cells-09-02355-f004:**
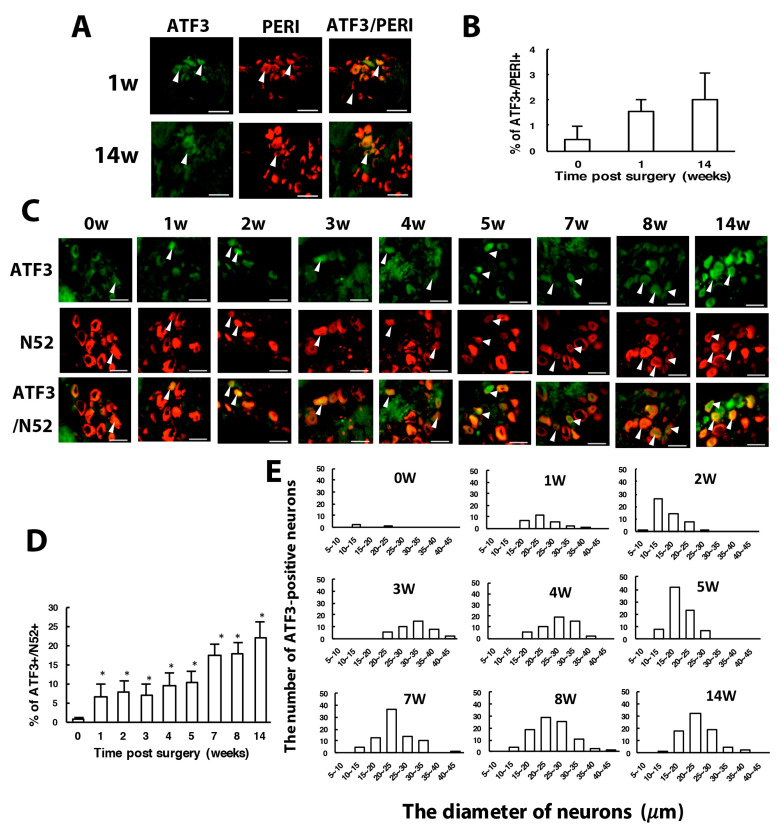
The number of ATF3-positive neurons is increased with time after CCI surgery. Ipsilateral L5 DRG were excised at 0, 1, 2, 3, 4, 8, 14 w after CCI surgery and co-stained with anti-ATF3 and anti-PERI antibodies (**A**,**B**) or anti-ATF3 and anti-N52 antibodies (**C**–**E**). (**A**) Neuron images showing green fluorescence for ATF3-positive (ATF3^+^) cells and red for PERI-positive (PERI^+^) cells. Arrowheads indicate ATF3^+^/PERI^+^ neurons. Scale bar is 50 μm. (**B**) Percentage of ATF3^+^/PERI^+^ neurons among PERI^+^ neurons (0 w, n = 3310; 1 w, n = 1107; 14 w, n = 637 PERI+ neurons). (**C**) Neuron images show green fluorescence for ATF3^+^ cells and red for N52-positive (N52^+^) cells. Arrowheads are ATF3^+^/N52^+^ neurons. Scale bar is 50 μm. (**D**) Percentage of ATF3^+^/N52^+^ neurons among N52^+^ neurons. Each time point contains n = 358-771 N52^+^ neurons. * *p* < 0.05. (**E**) Number of ATF3^+^/N52^+^ neurons in neuron groups of different diameters. 0 w, n = 3; 1 w, n = 26; 2 w, n = 50; 3 w, n = 41; 4 w, n = 51; 5 w, n = 80; 7 w, n = 79; 8 w, n = 92; 14 w, n = 79 ATF3^+^/N52^+^ neurons. n = 2 mice.

**Figure 5 cells-09-02355-f005:**
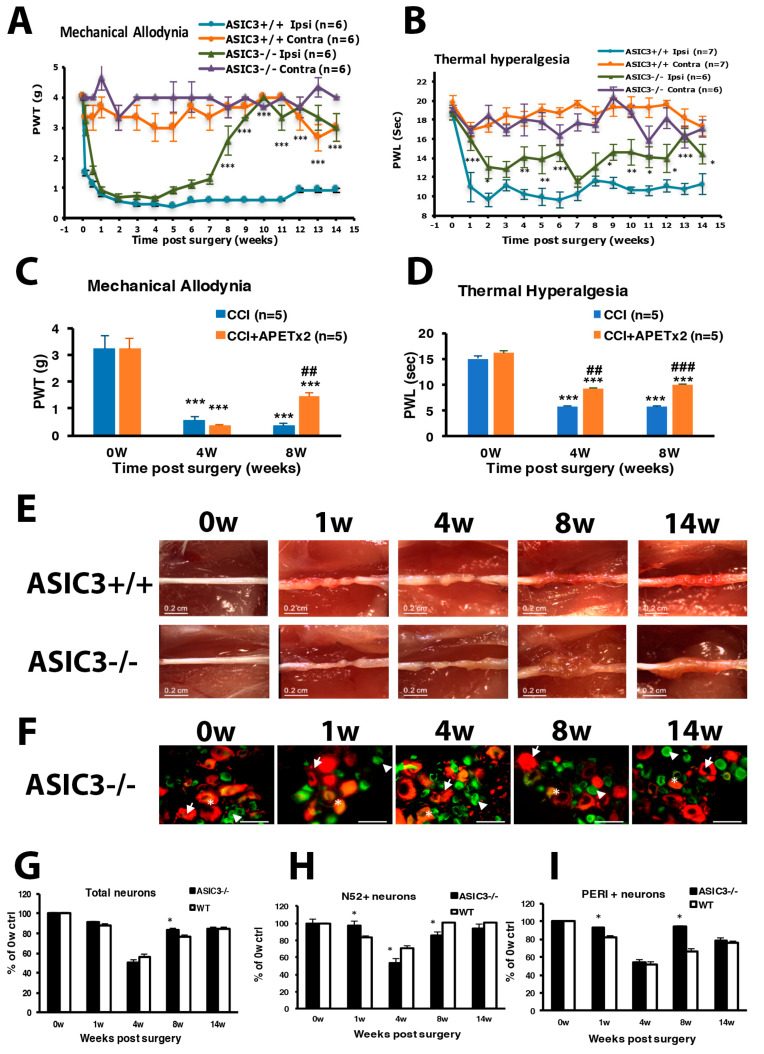
ASIC3 deletion shortens mechanical hyperalgesia and attenuates thermal hyperalgesia. The sciatic nerve of ASIC3^+/+^ or ASIC3^−/−^ mice was loosely ligated, then PWT (**A**) or PWL (**B**) was measured before (0 w) or at 1–14 w after surgery. Data are mean ± SEM of PWT or PWL (n = 6–8). * *p* < 0.05; ** *p* < 0.01; *** *p* < 0.001 for ASIC3^+/+^ ipsilateral vs. ASIC3^−/−^ ipsilateral by two-way ANOVA with post-hoc Bonferroni test. (**C**,**D**) Mice were intrathecally injected without or with 20 pmol of APETx2 before (0 w) or at 4, 8 w after CCI surgery. Mechanical (**C**) or thermal (**D**) tests were performed at 1 h after APETx2 injection. n = 5 mice. *** *p* < 0.001 for 4, 8 w vs. 0 w; ## *p* < 0.01; ### *p* < 0.001 for CCI+APETx2 vs. CCI. (**E**) The sciatic nerve from the ipsilateral side of ASIC3^+/+^ or ASIC3^−/−^ mice was excised at 0, 1, 4, 8, 14 weeks and the ligature was removed, followed by observation under the dissecting microscopy. Scale bar is 0.2 cm. (**F**) L5 DRG from the ipsilateral side of ASIC3^−/−^ mice were excised at 0, 1, 4, 8, 14 weeks after CCI surgery, then immunostained with anti-PERI (green) or anti-N52 (red) antibodies. Cell images are in D, and histograms show percentage of total neurons (**G**), N52^+^ neurons (**H**), and PERI^+^ neurons (**I**). * *p* < 0.05 for ASIC3^−/−^ vs. WT. Arrows indicate PERI^+^ neurons and arrowheads N52^+^ neurons. Scale bar is 50 μm. n = 872-1741 neurons; n = 2 mice.

**Figure 6 cells-09-02355-f006:**
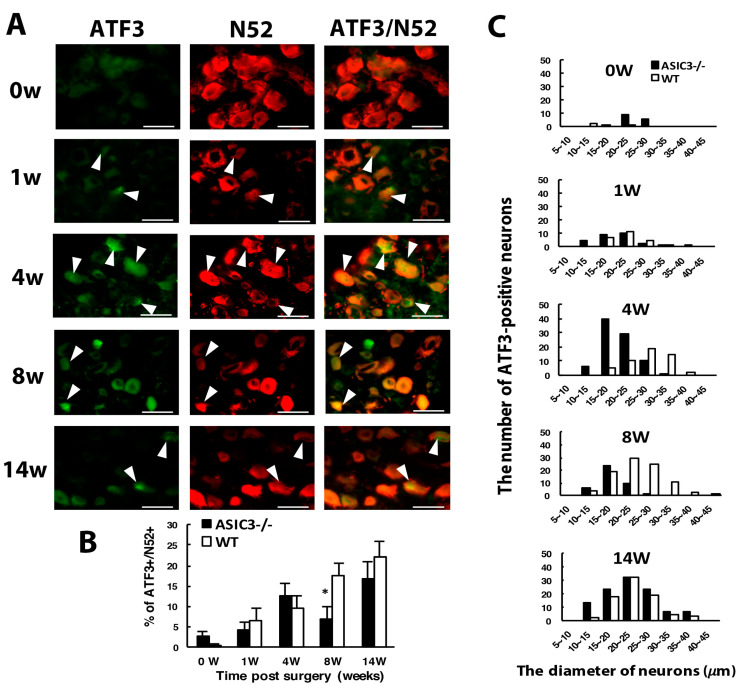
ASIC3 deletion shifts ATF3-positive neuron populations to small-diameter neurons. The ipsilateral L5 DRG was excised from ASIC3^−/−^ mice at 0, 1, 4, 8, 14 w after CCI surgery, and co-stained with anti-ATF3 and anti-N52 antibodies. (**A**) Neuron images show green fluorescence for ATF3^+^ cells and red for N52^+^ cells. Arrowheads indicate ATF3^+^/N52^+^ neurons. Scale bar is 50 μm. (**B**) Percentage of ATF3^+^/N52^+^ neurons among N52^+^ neurons. WT 0 w, n = 592; 1 w, n = 392; 4 w, n = 532; 8 w, n = 522; 14 w, n = 358 N52^+^ neurons; ASIC3^−/−^ 0 w, n = 552; 1 w, n = 682; 4 w, n = 677; 8 w, n = 609; 14 w, n = 641 N52^+^ neurons. * *p* < 0.05. (**C**) Number of ATF3^+^/N52^+^ neurons in neuron groups of different diameters. WT 0 w, n = 3; 1 w, n = 26; 4 w, n = 51; 8 w, n = 92; 14 w, n = 79 N52^+^ neurons; ASIC3^−/−^ 0 w, n = 16; 1 w, n = 29; 4 w, n = 86; 8 w, n = 42; 14 w, n = 108 N52^+^ ATF3^+^/N52^+^ neurons. n = 2 mice.

**Figure 7 cells-09-02355-f007:**
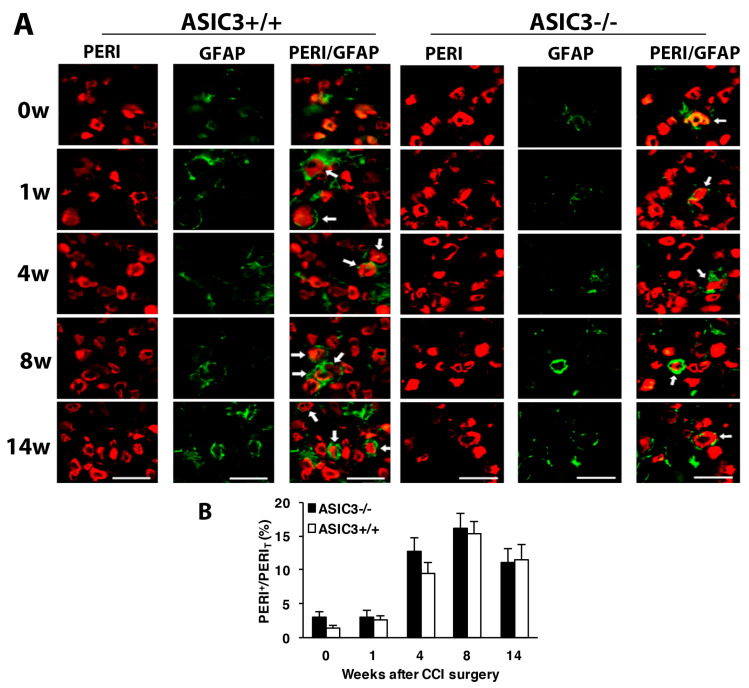
Number of satellite glial cells remains unchanged after ASIC3 deletion. The ipsilateral L5 DRG were excised from ASIC3^+/+^ or ASIC3^−/−^ mice at 0, 1, 4, 8, 14 w after CCI surgery and co-stained with anti-GFAP and anti-PERI antibodies. (**A**) Cell images showing GFAP (green) or PERI (red)-positive neurons. Scale bar is 50 μm. (**B**) Percentage of PERI-positive neurons surrounded by GFAP^+^ cells (PERI^GFAP+^) to total PERI-positive (PERI_T_) neurons. ASIC3^+/+^ n = 601–1315 neurons; ASIC3^−/−^ n = 797–1302 neurons, n = 2–3 mice.

**Figure 8 cells-09-02355-f008:**
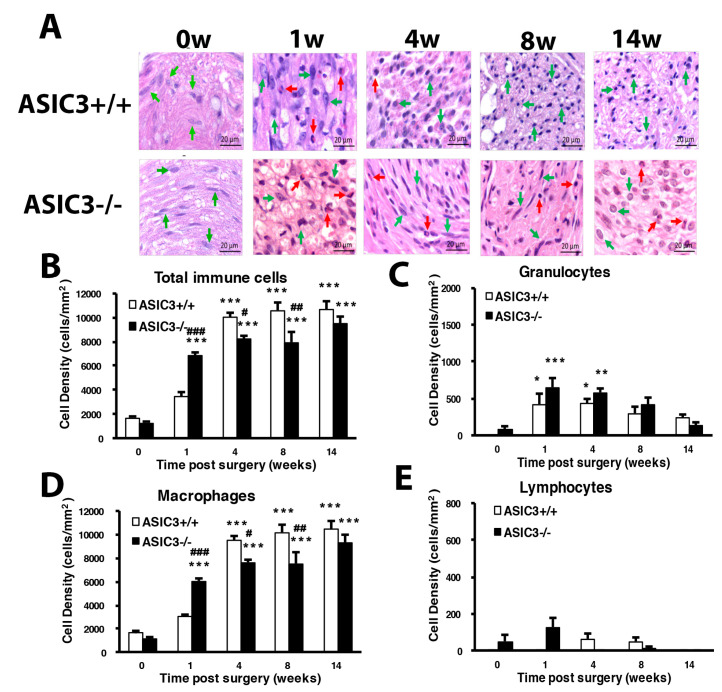
Mice lacking ASIC3 gene induce alteration of number of immune cells. The sciatic nerves from the ipsilateral side of ASIC3^+/+^ or ASIC3^−/−^ mice were excised before (0 w) or at 1, 4, 8, 14 w after CCI surgery, then fixed and cross-sectioned, followed by H and E staining. Representative H and E-stained sections showing infiltration of immune cells in the sciatic nerve (**A**). Scale bar is 20 μm. Histograms show the cell density of total immune cells (**B**), granulocytes (**C**), macrophages (**D**), and lymphocytes (**E**). Total cell number was calculated from regions of 1 mm^2^ to obtain cell density (n = 3 mice). # *p* < 0.05; ## *p* < 0.01; ### *p* < 0.001 for ASIC3^−/−^ vs. ASIC3^+/+^ by two-way ANOVA with post-hoc Bonferroni test.

**Figure 9 cells-09-02355-f009:**
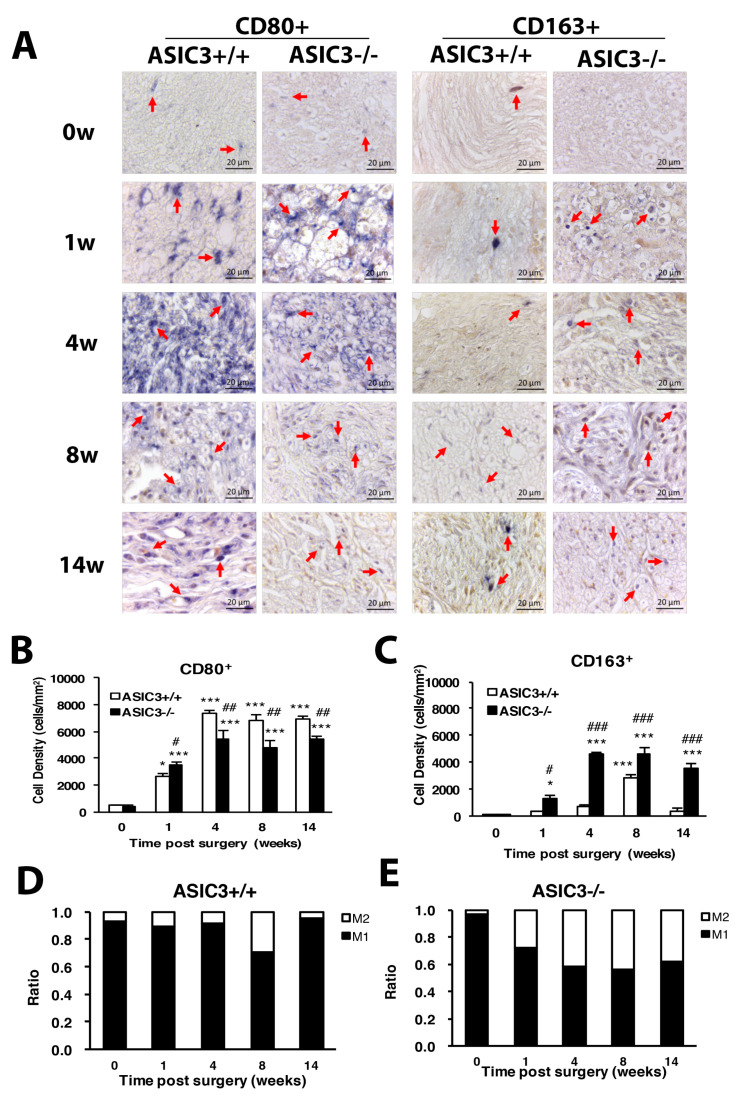
ASIC3 deletion increases M2 macrophage number but reduces M1 macrophage number. The sciatic nerves from the ipsilateral side of ASIC3^+/+^ or ASIC3^−/−^ mice were excised before (0 w) or at 1, 4, 8 and 14 w after CCI surgery, then fixed and cross-sectioned, followed by incubation with anti-CD80 (CD80^+^ for M1 macrophages) or anti-CD163 (CD163^+^ for M2 macrophages) antibodies. Representative sections showing macrophages in the sciatic nerve (**A**). Scale bar is 20 μm. Arrows indicate macrophages. (**B**,**C**) Cell density of CD80^+^ (M1) (**B**) or CD163^+^ (M2) (**C**) macrophages. Total cell number was calculated from regions of 1 mm^2^ to obtain the cell density (n = 3 mice). # *p* < 0.05; ## *p* < 0.01; ### *p* < 0.001 for ASIC3^−/−^ vs. ASIC3^+/+^ by two-way ANOVA with post-hoc Bonferroni test. (**D**,**E**) M1 and M2 ratio in ASI3^+/+^ (**D**) and ASIC3^−/−^ (**E**) sciatic nerve.

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
