# Peer review of "Deletion of Acid-Sensing Ion Channel 3 Relieves the Late Phase of Neuropathic Pain by Preventing Neuron Degeneration and Promoting Neuron Repair"

_cells, 2020, doi:10.3390/cells9112355_

Round 1
Reviewer 1 Report
The paper by Kun et al. provides novel knowledge on the role of ASIC3 channels in neuropathic pain. The authors use a chronic constriction injury of the sciatic nerve (CCI) in wild-type and ASIC3-/- mice. Behavioural tests and neuronal markers show that the absence of ASIC3 has an apparently protective effect. The authors suggest that lack of ASIC3 might prevent neuronal degeneration and macrophage polarization. Both actions would drive to the attenuation of neuropathic pain.
This is a descriptive study that lacks a mechanistic insight. The authors overinterpret their data and the discussion is too speculative. The paper should be more focused on providing robust data to better illustrate the importance of ASIC3 in neuropathic pain.
Major points:
- Number of observations. Line 95 of the methods section state that “Efforts were made to minimize the number of animals used and their suffering”. Figure legends do not indicate how many animals were used in the bar charts. This point is absolutely key. Data presentation must be substantially improved. All bar charts must be substituted by scatter plots to show the dispersion of data and the number of animals used in each experimental group should be indicated. Survival rates should also be indicated.
- Immunocytochemistry for ASIC3. The authors should apply immunostainings for this type of acid-sensing ion channel. ASIC3 -/- are the ideal control to find out if antibodies are working correctly. The paper only looks at the expression of ATF3 and N52. It is very interesting to see what is the population of neurons that express ASIC3. The authors must know what proportion of neurons express ASIC3. Obtaining this information would significantly improve the discussion and the significance of this work.
Minor points
- Behavioural tests. Use colour codes for the different experimental groups.
- Does it label all satellite cells or just a population? I recommend using other markers, as for example S100B to get a more accurate view of gliosis.
- Line 378.Oncreased.
Reviewer 2 Report
Chia-Chi Kung and collaborators examined whether a voltage-insensitive proton-gated cation channel ASIC3 is involved in neuropathic pain occurring after nerve injury. The experiments were performed by applying the behavioral, histological and immunohistochemical technique to chronic constriction injury (CCI) mice model. As a result, they found out that CCI mice exhibit persistent hyperalgesia for mechanical and thermal stimuli, long-lasting sciatic nerve inflammation, DRG neuron degeneration and increased expression of satellite glial cells (SCGs) and activating transcription factor 3 (ATF3). Furthermore, ASIC3 gene deletion was found to produce an inhibition of mechanical allodynia and themal hyperalgesia, a shift of AFT3 expression from large to small DRG neurons and an alteration of M1/M2 macrophage ratio (M1 number decrease and M2 number increase). It was suggested that ASIC3 deletion inhibits neuropathic pain by neuronal degeneration depression and neuronal repair promotion. Although the results obtained appear to be interesting, there are several points that should be addressed and may serve to amend this manuscript, as follows:
Major points:
- Lines 76 and 77: please introduce shortly suggested mechanisms for an involvement of ASIC3 in acute or chronic pain in published papers.
- Line 89: were the mice used a mix of male and female throughout experiments? Was there a difference in results between male and female mice? It is well-known that there is a difference in pain sensitivity between male and female animals (for example, see J Pain 2012 December; 13(12): 1224–1231. doi:10.1016/j.jpain.2012.09.009). Please address to this question.
- 2-4, 6-9, and lines 173, 174, 190 and 191: the authors should give experimental number for each of data having SEM.
- Line 331: Figure 7 does not have (C). Please amend this point.
- Page 15: please mention shortly why ATF3 expression results in prevention of neuronal degeneration.
- Although the authors seem to consider that ASIC3 mediates macrophage maturation produced by acidosis, what are possible mechanisms for this maturation? Is it possible that ASIC3 antagonists inhibit neuropathic pain? Are there candidates of the antagonists? Please discuss these points.
Minor points:
- Line 7: not “city” but “City”.
- Line 44: not “[TNFa]” but “(TNFa)”.
- Line 64: please define here CREB, because CREB is used in line 387 with no definition.
- Line 120: not “Trition-X” but “Triton-X”.
- Lines 171 and 172: it is unnecessary to define PWT and PWL, because they are defined in lines 153 and 158.
- Line 184: is “0.5 cm” right? Figure 1C presents “0.2 cm”. Please check this point.
- Line 262: not “are” but “indicate”.
- Line 314: not “were” but “was”.
- Line 316: not “are” but “indicate”.
- Line 406: please correct English.
- Please correct references 16, 26, 28 and 45. The other references should be checked.
Round 2
Reviewer 1 Report
The authors have addressed my concerns and I recommend publication in Cells.